# Impact of Alarm Fatigue on the Work of Nurses in an Intensive Care Environment—A Systematic Review

**DOI:** 10.3390/ijerph17228409

**Published:** 2020-11-13

**Authors:** Katarzyna Lewandowska, Magdalena Weisbrot, Aleksandra Cieloszyk, Wioletta Mędrzycka-Dąbrowska, Sabina Krupa, Dorota Ozga

**Affiliations:** 1Department of Anaesthesiology Nursing & Intensive Care, Medical University in Gdansk, 80211 Gdansk, Poland; kalewandowska@gumed.edu.pl; 2Intensive Care Unit, University Clinical Centre in Gdansk, 80211 Gdansk, Poland; magweisbrot@gmail.com; 3Independent Team of Physiotherapists, University Clinical Centre in Gdansk, 80211 Gdansk, Poland; cieloszczykaleksandra@gmail.com; 4Departament of Emergency, Institute of Health Sciences Medical College of Rzeszow University, 35310 Rzeszow, Poland; sabinakrupa@o2.pl (S.K.); gdozga@poczta.fm (D.O.)

**Keywords:** alarm fatigue, clinical alarms, critical care nurse, patient monitoring, patient safety

## Abstract

Background: In conditions of intensive therapy, where the patients treated are in a critical condition, alarms are omnipresent. Nurses, as they spend most of their time with patients, monitoring their condition 24 h, are particularly exposed to so-called alarm fatigue. The purpose of this study is to review the literature available on the perception of clinical alarms by nursing personnel and its impact on work in the ICU environment. Methods: A systematic review of the literature was carried out according to the guidelines of the Preferred Reporting Items for Systematic Reviews and Meta-Analysis (PRISMA) protocol. The content of electronic databases was searched through, i.e., PubMed, OVID, EBSCO, ProQuest Nursery, and Cochrane Library. The keywords used in the search included: “intensive care unit,” “nurse,” “alarm fatigue,” “workload,” and “clinical alarm.” The review also covered studies carried out among nurses employed at an adult intensive care unit. Finally, seven publications were taken into consideration. Data were analyzed both descriptively and quantitatively, calculating a weighted average for specific synthetized data. Results: In the analyzed studies, 389 nurses were tested, working in different intensive care units. Two studies were based on a quality model, while the other five described the problem of alarms in terms of quantity, based on the HTF (Healthcare Technology Foundation) questionnaire. Intensive care nurses think that alarms are burdensome and too frequent, interfering with caring for patients and causing reduced trust in alarm systems. They feel overburdened with an excessive amount of duties and a continuous wave of alarms. Having to operate modern equipment, which is becoming more and more advanced, takes time that nurses would prefer to dedicate to their patients. There is no clear system for managing the alarms of monitoring devices. Conclusion: Alarm fatigue may have serious consequences, both for patients and for nursing personnel. It is necessary to introduce a strategy of alarm management and for measuring the alarm fatigue level.

## 1. Background

In the conditions of an intensive care unit (ICU), where the patients cared for are in a critical condition, alarms are omnipresent [1]. Their purpose is to inform medical personnel of changes that occur in life parameters as well as of any failures of equipment. Sound signals may be generated by monitors, respirators, injection pumps, and many other devices. It turns out, however, that invariably such signals are false or clinically insignificant, amounting to even 85–99% of all alarms [2]. In spite of the fact that alarms form an important element of care and, in theory, they are designed to provide a patient with safety, their continuous onslaught may be overwhelming [3]. Nursing staff, who spend most of their time with patients, monitoring their condition 24 h, are particularly exposed to so-called alarm fatigue. The average number of alarms generated per patient, to which a nurse reacts when on duty, is from 150 to 400 [4]. Reacting to alarms constitutes 35% of the working time of a nurse in an ICU [5]. The sensory overburden caused by such an excessive amount of alarms may lead to delayed reactions to alarms or to ignoring them completely [6]. 

### 1.1. New Technologies—New Challenges for the Nursing Staff

The ECRI (Emergency Care Research Institute) is a global international organization that publishes an annual ranking of the most important hazards caused by medical technology. For many years, their list included the subject of alarm hazards. This hazard has been described as the lack of an adequate reaction to an alarm and poor management of alarms or their settings [7]. However, only in 2020 was the subject of alarm fatigue undertaken. Today, more than ever before, nursing personnel have to divide their attention between caring for patients and reacting to signals from numerous medical devices [8]. About 40 years ago, the number of alarms per patient in a critical condition was less than six. At present, there are more than forty such alarms [9,10]. Modern technology, with the introduction of more and more refined devices, makes the hospital environment even louder, while the sense of safety, which technology should theoretically guarantee, turns out to be illusory [11]. 

### 1.2. Nursing Staff Overload—Causes and Effects

Fatigue can be defined as a lack of energy to act. It can be acute, passing after a rest period, or chronic, characterized by irreversible physical and mental exhaustion [12]. In the case of alarm fatigue, it is defined as an excessive exposure to the stimulus generated by the monitoring unit [13]. Researchers highlight the importance of differentiation between acute and chronic fatigue; however, the literature on the subject does not clearly state which form the alarm fatigue is best categorized as [12]. Cognitive work overload relates to the need of prompt information processing by the nurses, and physical overload relates to the direct physical effort that takes place during the work [14]. Alarms cause work overload in quantity, quality, cognitive, and physical areas [15]. A big factor contributing to tiredness and alarm fatigue is noise. 

According to the guidelines of the World Health Organization, noise in the hospital environment should not exceed 35 dB. Noise disturbs communication and increases stress levels among personnel [16]. Unfortunately, such standards are very often exceeded. Konkani et al. proved in their project that the noise level calculated in an ICU was within 47–77 dB, and among the most significant sources of noise were false alarms [10]. The most common and best documented subjective reaction to noise is irritation, which may also include fear and mild anger, related to the conviction that one is interrupted on purpose [17]. A loud environment may discourage collaboration, intensify aggression, and hinder the capacity to process social signals [17]. Noise is a factor that may contribute to a sense of fear and stress in a ward. On the other hand, personnel work better in an environment that provides them with a sense of safety, peace, and space [18]. Prolonged exposure to this sensory factor results in stress, which is a determinant of occupational burnout in intensive care nurses [19]. Studies have shown that people adapt to noisy work environments, engage less interpersonally and reflectively, and show less concern for the patient and environment [14].

The causes of fatigue from monitoring device alarms vary. The necessity to recognize the alarm and assess and confirm its source is burdensome [20]. Under ICU conditions, many monitoring devices generate alarms of different priority. Alarms cause cognitive stress among employees, which is mainly caused by a break in the activity performed and the distraction from it and the prioritization of the urgency of the alarm [20].

Intensive care nurses feel irritation due to burdensome and false alarms every day, which generates a natural reaction in the form of subduing them or turning them off completely. As a result, important signals that require intervention may be ignored [21]. Malfunction, misuse, or inappropriate alarm setting of the monitoring devices may be harmful to the patient [22]. Between January 2009 and June 2012, 98 adverse events were recorded in the United States due to an incorrect or delayed reaction to an alarm, including 80 that ended in the death of a patient [2]. Alarm fatigue is a risk to patient safety. A case is described in which the medical personnel did not respond to the low-heart-rate alarm, which in consequence resulted in the patient’s death. In the investigation following this incident, the Centers for Medicare and Medicaid Services reported: “Nurses not recalling hearing low-heart-rate alarms were indicative of alarm fatigue, which contributed to the patient’s death” [23]. Attempting to shorten the alarm exposure of monitoring devices additionally leads to silencing of alarms in the monitoring center, without directly checking the patient’s condition [23].

Current literature on alarm fatigue has three major limitations to be addressed. First, maintaining a safe hospital environment, both for patient and staff. Second, the dynamic development of technology. Third, the dynamically changing environment of the intensive care unit. The purpose of the study is to review the available literature on the perception of clinical alarms by nursing staff in the intensive care unit. This paper is a current synthesis of the scientific evidence in relation to alarm fatigue by ICU nurses.

## 2. Method

### 2.1. Research Protocol and Guidelines

The systematic review of literature was carried out according to the PRISMA (Preferred Reporting Items for Systematic Reviews and Meta-Analysis) guidelines. The content of PubMed, OVID, EBSCO (electronic databases), ProQuest Nursery, and Cochrane Library was searched. A written protocol of the review was not drafted.

### 2.2. Search Strategy

The keywords used in the literature review were as follows: “intensive care unit,” “nurse,” “alarm fatigue,” “workload with nurse,” and “clinical alarm.” In the course of the search, single words were used or their combinations with AND/OR or both. The number of articles obtained during every search test was limited to research carried out in 2010–2019. Strict inclusion and exclusion criteria were applied. Seven publications were included in the study altogether. The detailed search process is presented in Figure 1. 

### 2.3. Study Selection

Inclusion criteria:▪articles in English,▪research group consisting only of medical personnel,▪research carried out among nurses working at adult intensive therapy units, to assess alarm fatigue among personnel, ▪studies with adult patients, and▪studies describing acute and chronic fatigue.

Exclusion criteria: ▪articles in a language other than English or Polish,▪research where a study group consisted of members other than medical personnel,▪moreover, articles were eliminated that focused on the perception of alarms generated by a single device (e.g., injection pumps or a pulse oximeter), and ▪studies concerning pediatric intensive therapy units.

### 2.4. Study Quality Assessment

All studies taken into account were assessed in terms of strength of evidence according to the Oxford Center for Evidence-Based Medicine 2011 Levels of Evidence [24].

### 2.5. Data Extraction 

Two reviewers assessed the studies independently, using a formalized form of data collection, which included, but was not limited to, the following data: the first author, the year of publication, the place of study, the study group, the type of study, and the method of assessing the perception of clinical alarms. Any and all disagreements were resolved by means of consensus and in consultation with another author. Seven articles were included in the descriptive analysis. In addition, quantity data from the abovementioned articles were synthetized and analyzed by another researcher, who managed to separate four articles whose shared values were subjected to a statistical analysis.

### 2.6. Data Analysis

Descriptive data were presented in the form of a table showing: the author and the year of publication, the country of study, the ward of study, the study group, the type of research, the method of assessing alarm fatigue, and the conclusion (Table 1).

Quantity data were analyzed based on the HTF (Healthcare Technology Foundation) study questionnaire. The abovementioned questionnaire was applied to four articles. Findings that describe the importance of clinical alarms were assessed using a five-point Likert scale with nine positions, in order establish the hierarchy of importance of barriers regarding the correct recognition of and reaction to alarms. Ranking statements on issues that inhibit the effective management of clinical alarms (Most important = 1 to Least important = 9) were calculated for the four articles (average values). For the needs of this study and in order to strengthen data, a weighted average was calculated from these results (Table 2). 

## 3. Results

Seven publications were qualified for the systematic literature review. The studies came from five different countries, including three from the United States [11,26,28], and one each from Australia [25], South Korea [9], Ireland [27], and Germany [29]. In all, 389 nurses were studied from the following units: ICU (intensive care unit), CCU (coronary care unit), TCICU (transplant/cardiac intensive care unit), HDU (high-dependency unit), PCU (progressive care unit), and PACU (post-anesthetic care unit). Two studies were carried out using a semi-structured interview [28,29]; five had the form of a descriptive, cross-sectional survey [9,11,25,26,27]; including four using the same HTF questionnaire study tool [9,11,26,27] and one a questionnaire created by the author [25]. 

### 3.1. Burdensome and False Alarms

In studies carried out by Christensen et al., 59% of questioned nurses reported that the inconvenience of alarms results from incorrectly set alarm thresholds [25]. In addition, 95% of nurses declared that they often felt the burden of alarms [26]. The inconvenience of alarms causes disturbances in the care of patients [9,11,26,27]. Importantly, the nuisance and falsity of alarms result in a reduction of trust in monitoring systems [11]. As a consequence of the large number of false alarms, nursing personnel are not capable of reacting to them in the proper manner [27]. According to the studied nurses, modern technologies are too complicated, while false alarms are too frequent and distract their attention [26].

### 3.2. Alarm Fatigue

According to 93% of nurses, alarm fatigue may cause alarms to be excessively subdued or ignored. In the same study, as many as 81% of respondents stated that alarm fatigue results from the excessive number of false alarms [25]. It is noteworthy that 52% of nurses do not know how to prevent alarm fatigue. Some of them declare that the only way is to adapt the alarms of devices showing patients’ life parameters to their health condition [27].

### 3.3. Alarm Management

With regard to the main obstacle to alarm management, the results of the studies are not unanimous. According to Sowan et al., nurses consider the difficulty in recognizing the source and priority of an alarm to be the main barrier [26], where the average weighted value obtained from the synthesis of data gives the result $\tilde{x}$ = 3.55. An additional barrier is the insufficient number of personnel, which prevents adequate reactions to alarms [11]. Among the least important obstacles to reacting to alarms, respondents listed: no training in the use of alarm systems, difficulty in the proper setting of alarms, and other non-clinical sounds, which is not completely confirmed by the data synthesis $\tilde{x}$ = 4.87. 

Christensen et al. administered their own 10-element questionnaire among Australian nurses. More than 50% of respondents thought that tiresome alarms result from the precision and incorrect settings of devices. Moreover, nurses suggest that more than half of alarms result from the absence of nurses at a patient’s bedside. On the one hand, this caused irritation and ignorance among other nurses, but on the other hand, some of them showed a sense of professional co-responsibility and reacted to the alarm signals of someone else’s patient [25]. 

In the quality studies, nurses present a sense of responsibility for the correct and individualized setting of alarms [28,29]. However, an excessive number of duties very often makes alarm management a task of low priority. American nurses feel overburdened with work and do not want to bear an exclusive responsibility for reacting to alarms, expecting support from other team members [28]. German nurses, who are also open to the introduction of mobile technology, report difficulties with advanced technology and numerous concerns about its implementation [29].

### 3.4. Risk of Bias

Two quality and five quantity studies were included in the research. The results of the quality studies are the voice of healthcare personnel who assess alarm fatigue. They are a necessary element to describe the final results; however, they may cause bias, being of a different form to other articles.

## 4. Discussion

### 4.1. Theoretical Implications

In this literature review, the focus was on publications that present the opinions and feelings of nurses regarding clinical alarms. After the analysis of results from studies conducted based on the HTF questionnaire, a simple conclusion can be drawn. Nurses from different parts of the world agree that burdensome alarms occur too frequently, disturb their care of patients, and reduce their trust in alarm systems [9,11,26,27]. The number of alarms per patient during a nurse’s duty may reach 150–400 or even more, out of which a definite majority are false or clinically insignificant alarms [25]. In the study conducted by Cho et al., the number of alarms generated by an ICU over 48 h was 2184, 36.2% of which were significant, while the remaining 63.8% were false [9]. 

HTF (Healthcare Technology Foundation) is an organization whose aim is to promote the safe use of technologies in healthcare. In 2005–2006, it conducted a national online questionnaire concerning the perception of clinical alarms by medical personnel. The same study was repeated in 2011 and in 2016. The latter one shows best how frequent such tiresome alarms are, accompanied by an increasing number of adverse events due to clinical alarms. The number of nurses who thought that burdensome alarms are too frequent amounted to 81% in 2006, 76% in 2011, and 87% in 2016 [28,30]. As confirmed by the results of studies carried out by Ruppel et al., nurses are afraid of the introduction of new technologies, as they may be burdened with even more work, considering their limited time and resources. Being overburdened with numerous duties in caring for patients, they do not want to be solely responsible for reacting to alarms. The involvement of a whole team in managing clinical alarms might contribute to the reduction of excessive and tiresome alarms. Nurses complain that alarms are ignored by doctors and that nursing personnel are relied upon exclusively in order to verify the importance of an alarm [31].

On the other hand, in the quality study carried out by Poncette et al., in Germany, nurses thought that the introduction of additional technology, such as tablets or mobile phones, might improve patient safety. Owing to the ability to cancel clinically irrelevant alarms from any location, stress might be reduced and satisfaction with performed work might be increased [29]. Nurses from a TCICU, in U.S. (Sowan et al.), who already work with mobile devices, i.e., mobile phones and pagers, have the opposite opinion. They consider mobile devices to be unreliable, sometimes experiencing a delay or losing the signal [26]. As described in the studies by Ruppel et al., carried out in the United States, the use of modern technologies may be problematic for older nurses. On the other hand, nurses with advanced clinical knowledge and experience usually showed greater comfort in adjusting alarms, and the level of noise during their duties was lower. This was related to better knowledge of physiological changes, recognizing certain “types” of patients and the ability to forecast different situations [28]. Other conclusions were drawn in a study carried out in Ireland (Casey et al.). The long-term experience and education of nurses had little impact on the knowledge of how to prevent alarm fatigue [27].

In 2013, the AACN (American Association of Critical-Care Nurses) published guidelines concerning alarm management. One of the recommendations was induction and continuous training [32]. An essential role in alarm management should be played by the education of nurses and the implementation of individual nursing internships. Such dynamically developing technology requires training. Other essential recommendations of the AACN concern, among others, suitable preparation of the skin for the daily exchange of electrodes (ECG), the change of pulse oximeter sensors if necessary, monitoring only those patients who have clinical recommendations, and creating a team responsible for the alarm system [6,32]. In 2018, the same organization published a new protocol based on evidence and ready strategies in order to solve problems related to alarm management. Based on such guidelines, Turmell et al. carried out a study to assess the effectiveness of strategies introduced at an American hospital. It turned out that the number of clinical alarms fell by 30%. Both before and after the introduction of the strategy, nursing personnel and monitoring technicians assessed clinical alarms with the HTF questionnaire. After the management strategy was implemented, 12% fewer personnel considered that disturbing alarms occurred too often. It was also observed that the respondents began to be more conscious of the rules of alarm management and more frequently discussed the issues of monitoring patients with their co-workers [33].

Monitoring the condition of a patient is one of the basic duties of nursing personnel. The results of the quality studies confirm that nurses are aware of that duty and feel responsible for the proper adjustment of alarms. Unfortunately, factors such as the overburdening number of duties; the insufficient number of nursing personnel; fear related to previous negative experiences, knowledge, and skills; or the lack of general aptitude in technologies very significantly influence the correct setting of alarms or alarm management in general [26,27,28]. In 2017, Sown et al. carried out a study in which a list of competences regarding the operation of monitors at an ICU was created, and the skills of nursing personnel in this respect were assessed. It turns out that 3–40% of nurses reported that they had never heard of 27 basic monitor functions and did not use them. The total number of competences included 54 basic and 5 advanced functions. The author emphasizes that only regular training guarantees the safe and proper use of monitors and reduces alarm fatigue. Training should be comprehensive and encompass not only basic but also more advanced functions [7]. 

The abovementioned literature review does not show the level of alarm fatigue but makes it possible to gain an insight into how alarms are perceived by nursing personnel and how they affect the daily work with patients. Only in the study by Cho et al. was a simple seven-element tool created for the needs of the project, outside the HTF questionnaire, in order to assess alarm fatigue. The obtained results show that Korean nurses felt fatigue on a moderate or higher level—69.4%. However, the tool was not completely reliable [9]. An attempt at developing another questionnaire to assess alarm fatigue among nurses at an ICU was carried out in Iran. This time the tool proved to be valid and reliable. Despite positive results, scientists stress the need to conduct further studies with a larger number of respondents. Alarm fatigue, as it constitutes a serious hazard both to patients and nurses, should be investigated and assessed, and a reliable tool is necessary for this purpose [34]. 

Summarizing the analyzed studies, we can say that nurses are exposed to too many false alarms. Second, nurses are overwhelmed by the introduction of new technologies and a sense of ownership of monitoring systems without the support of medical staff. Third, many years of experience of the nursing staff allow for the recognition of dangerous situations with the patient, but it does not help to prevent fatigue with alarms. Finally, it is worth focusing on ongoing training for nurses to increase the level of knowledge about alarm management in ICU conditions.

### 4.2. Practical Implications

In practice, efforts should be made to develop common universal principles for alarm management in all ICU sites around the world. Due to the variety of equipment, each ICU should have procedures dedicated to each unit, including compulsory training for young nurses or people joining the profession. At each stage of education relating to the ICU, training programs should be supplemented with content around the development of new technologies, thus adapting to the global needs of ICU branches and the existing market needs. This is mainly true for the young generation of nurses joining the profession, who must not forget that ICUs are not only alarms but also the patient and that the alarm must not distract them from the patient’s problems.

### 4.3. Limitation

The main limitation of the study was its inability to pinpoint the type of fatigue caused by the alarms. There are no explicit literature records describing acute and chronic fatigue associated with alarms from monitoring devices. Another limitation was the small number of articles meeting the criteria, which forced the researchers to include both quantitative and quantitative studies in the review. An experienced research team made an attempt to systematize the data. It turned out to be problematic to match the appropriate tool to assess the quality of the studies included in the review due to their diversity. More research into alarm fatigue is needed. The above analysis showed that there are many gaps in this respect. Only global research by scientists around the world will allow guidelines to be developed based on scientific evidence.

## 5. Conclusions

(1)Alarms are unavoidable in intensive care units. The dynamic development of technology makes their number grow drastically, and this will undoubtedly increase in the future as well. Therefore, it is necessary to introduce effective strategies of alarm management as soon as possible.(2)Nursing personnel feel overburdened with an excessive amount of duties and a continuous wave of clinical alarms.(3)Nurses often do not perceive the need for education regarding alarms, which is an important element of any alarm management strategy.(4)In the future, it is worth focusing on assessing the level of alarm fatigue. This would help provide safety both to patients and nursing personnel and verify the effectiveness of strategies that are introduced.

## Figures and Tables

**Figure 1 ijerph-17-08409-f001:**
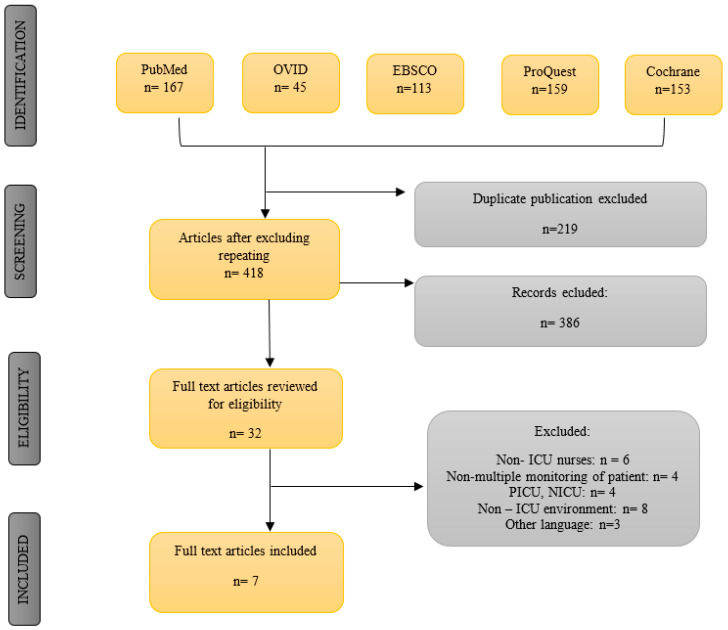
Scheme for articles qualified for a systematic review. ICU, intensive care unit; PICU, pediatric intensive care unit, NICU, neonatal intensive care unit.

**Table 1 ijerph-17-08409-t001:** Descriptive analysis of articles included in the systematic review.

Author and Year of Publication	Country of Study	Ward of Study	Study Group	Type of Research	Method of Assessing Alarm Fatigue	Conclusions
Christensen et al. (2014) [25]	Australia	ICU/CCU/HDU	48 nurses	Descriptive, survey design	Proprietary questionnaire: 8 open-ended questions 2 multiple choice questions	✓ 93% of respondents believe that fatigue caused by alarms can lead to silencing or ignoring them. ✓ 81% of nurses believe that fatigue caused by alarms is due to an excess of false alarms. ✓ 59% of nurses associate nuisance alarms with improperly set thresholds and alarm accuracy. ✓ 64% of nurses are aware that the correct setting of alarms should be based on the individual needs of the patient. ✓ surveyed nurses believe that over 50% of alarms are the result of the nurse’s absence at the patient’s bedside. ✓ 48% of nurses do not interfere with alarm settings of another nurse’s patient when an alarm occurs and she is absent.
Sowan et al. (2015) [26]	United States of America	TCICU	39 nurses	A cross-sectional survey Descriptive	Clinical study of alarms HTF (20 statements + 9 ranking positions)	✓ 98% of nurses say that nuisance alarms disrupt patient care and reduce confidence in alarm systems, inappropriately causing them to turn them off. ✓ 98% of nurses believe that nuisance alarms occur often. ✓ Surveyed nurses believe that difficulty in identifying the source and priority of an alarm is the most relevant cause disrupting alarm responses, the most irrelevant obstacle is the lack of training related to alarm systems. ✓ 22 nurses commented on the alarms. The main problems were as follows: false alarms are frequent and distracting, sound effects and visual indicators do not differ between the alarm’s priorities or parameters, modern technology is complex, remote monitoring (cell phones, pagers) is unreliable, it informs with a delay or not at all, there are no alarm management rules.
Cho et al. (2016) [9]	South Korea	ICU	77 nurses	A cross-sectional survey	Clinical study of alarms HTF (14 statements + 9 ranking positions)	✓ 94.8% of nurses believe that alarm sound effects and visual indicators should differ between priorities of alarms. ✓ 79.2% of nurses believe that nuisance alarms reduce trust in alarm systems, inappropriately causing them to turn them off. ✓ 76.6% of nurses believe that nuisance alarms are common. ✓ 66.3% of nurses believe that nuisance alarms are disrupting patient care. ✓ Surveyed nurses believe that too many false alarms is the most relevant obstacle disrupting the response to them, the most irrelevant is difficulties in setting an alarm correctly.
Petersen et al. (2017) [11]	United States of America	ICU PCU	26 nurses	A cross-sectional survey	Clinical study of alarms HTF (20 statements + 9 ranking positions)	✓ 88% of nurses believe that nuisance alarms are frequent. ✓ 100% of nurses believe that nuisance alarms reduce trust in alarm systems, inappropriately causing them to turn them off. ✓ 96% of nurses believe that nuisance alarms interfere with patient care and just as many believe that alarm sound effects and visual indicators should differ between priorities of alarms. ✓ 31% of nurses confirm that adverse events related to clinical alarms have occurred in a given facility in the last 2 years. ✓ Although 58% of nurses believe alarm management procedures are in place, only 35% of them are aware that they have a responsibility to document personalized alarm settings. ✓ Surveyed nurses believe that insufficient staffing is the most relevant obstacle disrupting the response to alarms, the most irrelevant is the sound of other non-clinical alarms and pagers.
Casey et al. (2018) [27]	Ireland	ICU PACU HDU	166 nurses	A cross-sectional survey	Clinical study of alarms HTF (19 statements + 9 ranking positions)	✓ 90% of nurses believe that nuisance alarms are common. ✓ 91% of nurses believe that nuisance alarms are disrupting patient care. ✓ 81% of nurses believe that nuisance alarms reduce trust in alarm systems, inappropriately causing them to turn them off. ✓ 89% of nurses say that they always adjust the alarm thresholds at the beginning of the shift and modify them accordingly during the day. ✓ 54% of nurses are aware of adverse events related to clinical alarms in their workplace. ✓ Surveyed nurses believe that too many alarms is the most relevant obstacle disrupting the response to alarms, the most irrelevant is the sound of other non-clinical alarms and pagers.
Ruppel at al. (2019) [28]	United States of America	ICU	27 nurses	Quality study	Semi-structured interviews	✓ The nurses agree that it is their responsibility to set alarm thresholds and, for most, checking for alarms at the beginning of their shift has become a habit. ✓ It has been observed that adjusting the alarms is related to the knowledge, skills, education, and "style" of the nurse. More experienced nurses have more freedom in setting alarms. ✓ Nurses, despite feeling obliged to manage alarms, do not want to be solely responsible for responding to alarms. They expect support from other team members. ✓ Nurses are often overwhelmed with other patient care responsibilities, making alarm management a low-priority task. ✓ Nurses have different motivations to set alarms. The external factor that motivates new nurses is so-called "Emergency police" (older, more experienced nurses). Others have an intrinsic, personal need to provide the best possible care to the patient caused by the fear of repeating errors from past situations.
Poncette et al. (2019) [29]	Germany	ICU	6 nurses	Qualitative Study	Semi-structured interviews	✓ Nurses say they regularly adjust alarm thresholds to meet patients’ needs. However, they have difficulty handling the advanced features of the monitor. ✓ Nurses considered fatigue with alarms, which manifests in turning all of them off, as a potential danger for the patient. ✓ Nursing staff believe that remote monitoring via mobile phones and tablets can increase patient safety, reduce hospital admission time in the ICU, and increase job satisfaction. ✓ Nurses identified obstacles caused by implementing innovative technologies as: lack of full trust in them, fear of more responsibilities with already limited resources and time, risk of reduced contact with the patient, and loss of clinical skills, lack of general awareness of current technologies.

ICU—intensive care unit, CCU—coronary care unit, HDU—high-dependency unit, TCICU—transplant/cardiacintensive care unit, PCU—progressive care unit, PACU—post-anesthesia care unit. HTF—Healthcare Technology Foundation.

**Table 2 ijerph-17-08409-t002:** Quantitative analysis, weighted average, and ranking statements on issues that inhibit effective management of clinical alarms.

Ranking Statements on Issues That Inhibit Effective Management of Clinical Alarms:		Sowan et al. (2015) [26]	Cho et al. (2016) [9]	Petersen et al. (2017) [11]	Casey et al. (2018) [27]	Weighted Average $\tilde{x}$
Frequent false alarms, which lead to reduced attention or response to alarms when they occur	Average	4.15	2.75	3.83	2.43	2.84
	Quantity	39	77	26	166	
Difficulty in understanding the priority of an alarm	Average	3.06	3.53	3.48	3.69	3.55
	Quantity	39	77	26	166	
Inadequate staff to respond to alarms as they occur	Average	4.23	4.86	3.13	2.66	3.45
	Quantity	39	77	26	166	
Difficulty in hearing alarms when they occur	Average	3.93	4.94	4.83	3.8	4.18
	Quantity	39	77	26	166	
Difficulty in identifying the source of an alarm	Average	2.94	5.22	3.65	3.81	4.04
	Quantity	39	77	26	166	
Over reliance on alarms to call attention to patient problems	Average	4.77	5.35	4.87	3.4	4.18
	Quantity	39	77	26	166	
Noise competition from non-clinical alarms and pages	Average	4.45	5.74	6.04	4.1	4.73
	Quantity	39	77	26	166	
Lack of training on alarm systems	Average	6.6	6.21	4.83	3.86	4.87
	Quantity	39	77	26	166	
Difficulty in setting alarms properly	Average	4.44	6.39	4.25	4.14	5.02
	Quantity	39	77	26	166	

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
