# Peer review of "Impact of Alarm Fatigue on the Work of Nurses in an Intensive Care Environment—A Systematic Review"

_ijerph, 2020, doi:10.3390/ijerph17228409_

Round 1

Reviewer 1 Report

The Authors provide a systematic review following the PRISMA protocol on the impact of alarm fatigue on the work of nurses in intensive care units. The authors should make an effort to improve some points of the study:

INTRODUCTION:

- The Authors should provide a more extensive state of the art (with bibliographic citations) on the consequences of alarm fatigue in intensive care nurses. In specific terms, they should include the concepts and the current situation in the scientific literature of acute fatigue, chronic fatigue, workload, etc. associated with alarm fatigue.

- The objectives of the study should be clarified and the study should be justified.

METHODOLOGY:

- The concept of alarm fatigue used by the authors when carrying out this systematic review should be defined: acute fatigue, chronic fatigue, intershift recovery, workload, etc. and the bibliographic search and systematic review carried out accordingly.

- The bibliographic review to obtain the articles should also include the Cochrane Library, the Proquest Nursery, and the Cumulative Index to Nursing and Allied Health Literature (CINAHL), where more studies to be included in the systematic review can be found, or their exclusion should be justified.

- The keywords in the search for scientific articles should also include “workload with nurse” or the other items mentioned above, or their exclusion should be justified.

- After defining which area of alarm fatigue the study focuses on (a definition would be necessary), the other concepts associated with alarm fatigue that have not been studied should be indicated in the exclusion criteria.

- The scientific quality assessment of the studies in terms of bias risk using the Epidemiological Appraisal Instrument (EAI) should be included.

RESULTS:

- The tables should have a footer explaining the abbreviations used (HDU, CCU, etc).

- The citations of authors in the tables include their bibliographic reference number in the study.

Reviewer 2 Report

This is a very interesting and well done review of alarm fatigue in intensive care units. As authors point out, this is a theme that can have an important impact both on workers and their patients. Workers are not only fatigued by the alarms, this also represents noise exposure that can potentially disrupt biological mechanisms and lead to disease in nurses. Hopefully more research and interventions working with ergonomic experts will be carried out.
As an only minor comment, if there were no articles on this subject published in Polish, authors could omit this from the methodology or mention how many articles were found on this subject in this language. This is more for informative reasons and to have an idea of how studied this has been in Poland (despite not having any articles as part of the review). 

Reviewer 3 Report

Comment on

Impact of Alarm Fatigue on the Work of Nurses in an Intensive Care Environment – a Systematic Review. 

  1. The introduction can be improved. First, put forward directly your research theme: the alarm fatigue on the work of nurses. Second, state the importance of the issue of alarm fatigue. Third, point out the limitations of the study in the current research of alarm fatigue. Third, articulate how you will contribute to the current literature.
  2. I would suggest adding some propositions and argument for the positions as a form of literature review. For example,

Proposition 1; Hospital alarm management practices (e.g., training) are negatively related to alarm fatigue.

Proposition 2: Frequency of false alarm is positively related to nurses’ alarm fatigue.

Proposition 3: Nurses’ problem coping is negatively related to alarm fatigue.

Proposition 4: Frequency of false alarm is negatively related to patients’ satisfaction at hospital.

Proposition 5; Alarm fatigue is positively related to nurses’ well-being (H5a), their silence (H5b) and turnover intention (H5c).

  1. Results

You may offer some support to the propositions through analyzing prior studies.

  1. In you Discussion part, clearly indicate the theoretical implications of your study. How do managers learn from your review? Offer some practical implications for managers. Please use “First, ---; Second, ---; Third, ---“. What are the limitations of the study and direction of future research?
  2. Please reorganize your Discussion part as suggested. Please avoid bringing about new information in this part.

Round 2

Reviewer 1 Report

The authors have made an effort to improve the paper.

Author Response

The authors have made an effort to improve the paper.

Response 1: Thank you

Reviewer 3 Report

comments:

  1. I would suggest further updating your Introduction parts. In my previous review, I suggested that you directly put forward alarm fatigue as your research theme without starting with noise. You may also state your research problem more clearly. For example, "Current literature on alarm fatigue has three major limitations to be addressed. First, ---. Second, ---. Third, ---". The purpose of the study is ---. 
  2. It is important to respond to all the comments by the reviewer. Unfortunately, you seem to take no notice of the suggestion of adding some propositions in the Introduction as literature review. 
  3. In your Discussion, please add headers like "Theoretical contributions", " Practical implications", and "Limitations". It is easy for readers to follow you if you use "The first contribution is ---. The second contribution is ---".  

Author Response

I would suggest further updating your Introduction parts. In my previous review, I suggested that you directly put forward alarm fatigue as your research theme without starting with noise. You may also state your research problem more clearly. For example, "Current literature on alarm fatigue has three major limitations to be addressed. First, ---. Second, ---. Third, ---". The purpose of the study is

Response 1: Introductory part contains information describing the problem of alarm fatigue. Noise is a sensory overload resulting from alarm fatigue, which we emphasize in the introduction. That is why it is so important for us to show in this part the consequences of noise overload in the context of monitoring devices.

In order to meet your comments, we have specified the research problem and described the main limitations using terms as suggested [93-98].

It is important to respond to all the comments by the reviewer. Unfortunately, you seem to take no notice of the suggestion of adding some propositions in the Introduction as literature review.

Response 2: Forgive me if you have the impression that we are not following the comments. Your suggestions are key to getting the manuscript as good as possible.

In your Discussion, please add headers like "Theoretical contributions", " Practical implications", and "Limitations". It is easy for readers to follow you if you use "The first contribution is ---. The second contribution is ---".  

Response 3: Discussion headers in relation to your suggestions have been added. Thanks to this, the structure is clear. Additionally, some of the implications for practice and limitations have been expanded.

Round 3

Reviewer 3 Report

Comment

1.     In the Introduction part, you need to further clarify and highlight the major topic of your study: cause and effect of alarm fatigue.

2.      In your Discussion part, change theoretical contribution to theoretical implication. In this section, please summarize the major issues (cause and effect of alarm fatigue) that you find in reviewing this stream of research. Use “First, ---. Second, ---. Third, ---. Finally, ---”. Researchers can cite your work later in their studies if you can articulate more clearly the predictors and outcomes of alarm fatigue of nurses.

Author Response

In the Introduction part, you need to further clarify and highlight the major topic of your study: cause and effect of alarm fatigue.

Response: The elements indicated by the reviewer were added in the introduction. Causes[87-91];  Effects [98-104]. Changes in the text are marked in red.

In your Discussion part, change theoretical contribution to theoretical implication. In this section, please summarize the major issues (cause and effect of alarm fatigue) that you find in reviewing this stream of research. Use “First, ---. Second, ---. Third, ---. Finally, ---”. Researchers can cite your work later in their studies if you can articulate more clearly the predictors and outcomes of alarm fatigue of nurses.

Response: We changed theoretical contribution to theoretical implication [62]. We add summarize part [136-141]. Changes in the text are marked in red.